# The Role of Reirradiation in Childhood Progressive Diffuse Intrinsic Pontine Glioma (DIPG): An Ongoing Challenge beyond Radiobiology

**DOI:** 10.3390/brainsci13101449

**Published:** 2023-10-11

**Authors:** Maria Chiara Lo Greco, Roberto Milazzotto, Rocco Luca Emanuele Liardo, Pietro Valerio Foti, Stefano Palmucci, Antonio Basile, Stefano Pergolizzi, Corrado Spatola

**Affiliations:** 1Radiation Oncology Unit, Department of Biomedical, Dental and Morphological and Functional Imaging Sciences, University of Messina, 98122 Messina, Italy; stefano.pergolizzi@unime.it; 2Radiation Oncology Unit, Department of Medical, Surgical Sciences and Advanced Technologies “G.F. Ingrassia”, University of Catania, 95123 Catania, Italy; r.milazzotto@policlinico.unict.it (R.M.); lucaliardo@hotmail.com (R.L.E.L.); cor_spatola@hotmail.com (C.S.); 3Radiology I Unit, Department of Medical Surgical Sciences and Advanced Technologies “G.F. Ingrassia”, University of Catania, 95123 Catania, Italy; pietrofoti@hotmail.com (P.V.F.); spalmucci@unict.it (S.P.); basile.antonello73@gmail.com (A.B.)

**Keywords:** DIPG, high-grade glioma, midline gliomas, pediatrics, reirradiation, vinorelbine, nimotuzumab, radiobiology

## Abstract

To investigate the clinical impact of multiple courses of irradiation on pediatric patients with progressive diffuse intrinsic pontine glioma (DIPG), we conducted a retrospective case series on three children treated at our institution from 2018 to 2022. All children were candidates to receive systemic therapy with vinorelbine and nimotuzumab. Radiotherapy was administered to a total dose of 54 Gy. At any disease progression, our local tumor board evaluated the possibility of offering a new course of radiotherapy. To determine feasibility and assess toxicity rates, all children underwent clinical and hematological evaluation both during and after the treatment. To assess efficacy, all children performed contrast-enhanced MRI almost quarterly after the end of the treatment. In all children, following any treatment course, neurological improvement (>80%) was associated with a radiological response (41.7–46%). The longest overall survival (24 months) was observed in the child who underwent three courses of radiotherapy, without experiencing significant side effects. Even though it goes beyond the understanding of conventional radiobiology, first and second reirradiation in pediatric patients with progressive DIPG may represent a feasible and safe approach, capable of increasing overall survival and disease-free survival in selected patients and improving their quality of life.

## 1. Introduction

Diffuse intrinsic pontine glioma (DIPG) is the most common childhood brainstem malignancy, accounting for nearly 75–80% of all brainstem tumors in this age group [1]. Over the last few decades, the prognosis for DIPG has remained dismal due to the absence of effective therapies, resulting in a 2-year overall survival (OS) rate of less than 10% [2,3].

Due to its local infiltration and brainstem localization, this disease is not amenable to surgical resection, and the only available treatment modality is radiotherapy, traditionally delivered by conventionally fractionated radiotherapy (CFRT) to a total dose of 54 Gy [4].

Radiotherapy has been shown to achieve neurological improvement in nearly 70% of patients and MRI objective responses in 30–70% of patients. Unfortunately, within 3–8 months after the completion of the treatment, most children show clinical or radiographic evidence of disease progression [1,5]. In this setting, no standard of care exists and the only treatment option at our disposal is reirradiation.

To investigate the clinical impact of multiple courses of irradiation on pediatric patients with progressive DIPG, we conducted a retrospective study on three children treated at our institution with systemic therapy and two or three courses of radiation therapy.

## 2. Materials and Methods

For this study, we retrospectively selected three children diagnosed with DIPG, treated at our institution from 2018 to 2022. Median age was 5 ± 1.4 years (3 patients).

At first clinical observation, all children presented neurological symptoms (such as ataxia, dysmetria, dysarthria, and cranial nerve dysfunction) suggestive of brainstem involvement. Subsequently, all children underwent neuroimaging with contrast-enhanced MRI. This examination revealed an intrinsic, pontine-based infiltrative lesion with indistinct borders. The lesions appeared to be hyperintense on T2-weighted and FLAIR sequences and hypointense on T1-weighted sequences, consistent with the infiltrative nature of DIPG. At baseline, the median diameter was 40 ± 4.2 mm (3 patients).

Histological confirmation was not performed due to the considerable risks associated with DIPG biopsy; therefore, the diagnosis was radiologically based.

According to our institutional protocol, all patients were candidates to receive systemic therapy with vinorelbine 20 mg/m^2^ (weekly administered) and nimotuzumab 150 mg/m^2^ (in the first 12 weeks of treatment). Radiotherapy was delivered from weeks 3 to 9, for a total dose of 54 Gy. Vinorelbine 25 mg/m^2^ and nimotuzumab were given every other week thereafter until the tumor progressed or for up to 2 years.

The choice to pursue this treatment strategy is based on the promising results of a non-randomized, open-label phase II pilot study published in 2014. In this study, the authors evaluated the efficacy, in terms of objective response rate, of combining nimotuzumab and vinorelbine with radiation therapy (54 Gy delivered in 1.8 Gy daily fractions) in newly diagnosed DIPG [6].

To design an adequate and personalized treatment plan, all children underwent a radiotherapy simulation process, under anesthesia, through a brain CT scan without contrast enhancement. Since the patient’s position needed to be the same for both the initial simulation and subsequent treatment, an accurate set-up was needed. During this process, the children were placed in the supine position with their heads immobilized using a custom-made thermoplastic mask. CT images were acquired with a 2 mm slice spacing, from the cranial vertex to C7, and then data were analyzed using Treatment Planning Software (Monaco^®^, version 5.11) for target volume definition and dose solutions, according to our institutional protocol.

During the contour delineation phase, images were co-registered with T2-weighted and gadolinium-enhanced T1-weighted MRI sequences to precisely identify the gross tumor volume (GTV). Subsequently, the clinical target volume (CTV) and the planning target volume (PTV) were created with a total expansion of about 2.0 cm from the GTV to the PTV. The treating radiation oncologist outlined the relevant organs at risk (OARs), including the brainstem, optic nerves, optic chiasm, and spinal cord. Efforts were made to develop a homogenous conformal dose distribution and minimize the dose to surrounding uninvolved structures.

The first course of radiation therapy was delivered using intensity-modulated radiotherapy (IMRT) with a 6 MV linear accelerator (Siemens, ONCOR), concurrent with dexamethasone (8 mg daily). Anesthesia was used as needed for all treatment delivery sessions.

At disease progression, our local tumor board evaluated the possibility of offering a new radiotherapy course instead of palliative care. This decision was made taking into account several factors, such as the potential clinical benefits, the risk of adverse events, the life expectancy of the children, and the will of the family.

In this context, simulation, contouring, and planning procedures were much the same as for the first-line treatment. The beam geometry for reirradiation was chosen to avoid the entrance beam paths of the first-line treatment wherever possible. As well as for the first course, reirradiation was administered alongside dexamethasone (8 mg daily).

To determine the feasibility and toxicity rates of the therapies, all children underwent clinical and hematological evaluations once a week throughout the entire treatment course, and quarterly after the end of it. To determine the efficacy of the therapies, patients underwent contrast-enhanced MRI almost quarterly after the end of the treatment.

## 3. Results

After radiological diagnosis, all children were candidates to start chemotherapy with nimotuzumab and vinorelbine, followed, after 3 weeks, by CFRT to a total dose of 54 Gy, delivered in daily fractions of 2 Gy.

For the first radiotherapy course, the biologically effective dose (BED) was calculated using the following formula: BED = nd (1 + d/(α/β)), where n is the number of fractions, d is the dose per fraction, and α/β stands for alpha beta ratio.

According to radiobiology knowledge, early-responding tissues with rapid turnover are characterized by a high α/β ratio, while late-responding tissues with slow turnover are characterized by a low α/β ratio. For this reason, we decided to use an α/β value of 10 Gy (BED10) for tumor effects and an α/β value of 3 Gy (BED3) for late effects. Therefore, we quantified BED10 = 64.8 Gy, and BED3 = 90 Gy [7,8].

During the first treatment course, all patients gradually manifested clinical improvements, showing neither neurological deficits nor limitations of physical and mental performance. The nimotuzumab-vinorelbine combination was very well tolerated. All children showed mild adverse events, including mood swings, irritability, vomiting, and sleep disturbance. No severe adverse events were shown.

Two months after the end of the first radiation therapy course, MRI imaging was performed, showing a notable reduction in the neoplastic mass in all three patients, with the median diameter decreased to 21.6 ± 2.5 mm.

The median time to first progression was 7 ± 1.7 months (three patients). Then, new symptoms appeared, consistently with radiological progression (the median diameter increased in all three patients to 34.3 ± 2.5 cm^3^). Hence, our local tumor board decided to offer a second course of CFRT to all children, with a total dose of 19.8 Gy delivered in daily fractions of 1.8 Gy. Since it is universally recognized that the planning protocol for reirradiation is much more complex than that for primary treatment, it was considered necessary to quantify dose–volume records from both radiations, and to consider the time lapse between the two regimens and the regeneration capacity of OAR, as well as the tumor control probability versus normal tissue complication probability in each dose–volume histogram (DVH).

For the second radiotherapy course, we quantified BED10 = 23.4 Gy and BED3 = 31.7 Gy. The total cumulative BED (BEDxcum) was calculated using the following formula: BEDxcum = BEDxinitial + BEDx; so we quantified BED10cum = 88.2 Gy and BED3cum = 121.7 Gy.

In this setting, to obtain the reirradiated dose, the 2 Gy equivalent normalized dose (EQD2) was calculated using the following formula: EQD2 = BED/(1 + 2/(α/β)), where D is the total dose [8]. Then, the total cumulative EQD2 was calculated using the following formula: EQD2cum = EQD2initial + EQD2. Considering both α/β = 10 Gy and α/β = 3 Gy, we quantified EQD2cum = 73.5 Gy and 73 Gy, respectively.

In all three patients, after transitory clinical and radiological benefits (the median diameter decreased to 20 ± 4 mm), the median time to the second progression was 4.3 ± 1.5 months. At this point, both neurological symptoms and radiological progression occurred, with the median diameter measuring 33.7 ± 2.8 mm (three patients). In this case, our tumor board decided to offer a third course of radiation therapy only to the child with the longest time to first and second progression (9 and 6 months, respectively) and ensure that the others would receive the best palliative care. The third course of CFRT was delivered to a total dose of 12 Gy in daily fractions of 2 Gy.

For the third radiotherapy course, we quantified BED10 = 14.4 Gy, with cumulative BED10 = 102.6 Gy, and BED3 = 20 Gy, with cumulative BED3 = 141.7 Gy; for α/β = 10 Gy and α/β = 3 Gy, EQD2cum = 85.5 Gy and 85 Gy, respectively.

In all children, after any radiotherapy course, neurological improvement was associated with radiological response, but since the clinical response (>80%) was higher than the radiological response (41.7–46%), no clear correlation between these two parameters could be claimed. The median OS was 19.6 ± 3.7 months, with the highest OS (24 months) observed in the child with the longest time to first and second progression who had the chance to be treated with three courses or radiotherapy (Figure 1 and Figure 2). The patients’ time to first and second progression and overall survival are reported in Table 1.

## 4. Discussion

### 4.1. Systemic Therapies

Despite collaborative efforts in designing prospective trials, studies investigating chemotherapy regimens in DIPG have been sparse due to the rarity of this disease and the uncertainty surrounding the histological diagnosis.

With the purpose of identifying the most efficient treatment, in 2006 Hargrave et al. published a comprehensive review of 29 clinical studies on DIPG, conducted from 1984 to 2005. This report revealed disheartening results in terms of survival advantage, with no systemic therapies showing any benefit over conventional radiotherapy [2].

Subsequently, a prospective trial investigating a pre-radiation chemotherapy regimen based on hematotoxic and non-hematotoxic courses was published. This study reported impressive results in terms of OS. Unfortunately, the survival advantage achieved with the administration of tamoxifen, BCNU, cisplatin, and high dose methotrexate, was obtained at the expense of a doubled period of hospitalization, significantly impacting patients’ quality of life [9].

Years later, based on the promising results provided by the Stupp protocol to treat glioblastoma [10], The Children’s Oncology Group (ACNS0126) decided to investigate the use of temozolomide concurrent to radiotherapy, followed by adjuvant temozolomide in DIPG patients. The study reported a 1-year OS non-significantly improved compared to the historical control [11].

Similarly, the use of capecitabine concurrent to radiotherapy in children with DIPG was investigated by The Pediatric Brain Tumor Consortium, but no improvements in terms of progression-free survival and OS were reported [12].

Since cytotoxic chemotherapies have historically been demonstrated to be ineffective, in recent years, to conduct biology-driven translational research supporting precision medicine, the role of biopsy has been redefined. In daily practice, histological confirmation is not mandatory and biopsy is currently indicated only in case of atypical radiological features or in the context of a clinical trial [13]. Nevertheless, research has revealed that the operative risk of brainstem biopsies with modern surgical techniques in specialist pediatric centers has progressively decreased, supporting the execution of this procedure for the detection of molecular alterations and the administration of targeted therapies [14].

In this context, the most common alterations in DIPG samples have been found to involve PDGFRA, EGFR, H3 (H3.1 or H3.3 K27M), TP53, PI3KCA, SOX2, OLIG2, MYCN, and ACVR1.

Given that one of the most frequently amplified genes in DIPG is PDGFRA, PDGFR-targeted therapies, including agents such as imatinib and dasatinib, have been investigated, unfortunately resulting in poor antitumor effects [15].

Another gene that is overexpressed in pediatric brain tumors is EGFR. To this purpose, clinical trials investigating anti-EGFR drugs including nimotuzumab, gefitinib, and erlotinib have been conducted, showing some benefits but only in small subsets of DIPG patients [16,17,18].

A pilot phase II study published in 2014 evaluated the efficacy of combining radiotherapy with nimotuzumab and vinorelbine, a semisynthetic vinca alkaloid capable of altering blood–brain barrier permeability and enhance the EGFR binding sites [6]. Since this study has shown encouraging results in terms of median PFS and OS (8.5 and 15 months, respectively), we decided to administer this treatment strategy to our patients [6].

Other trials have used PARP1 inhibitors (olaparib, niraparib, and veliparib), CDK4/CDK6 inhibitors (PD-0332991), WEE1 kinase inhibitor (MK1775), and the angiogenesis inhibitor (bevacizumab), but despite attempts, none of these have shown significant efficacy [19,20,21,22].

### 4.2. Radiotherapy

Nowadays, the cornerstone of treatment of newly diagnosed DIPG remains external beam radiotherapy, delivered in daily fractions of 1.8–2 Gy to a total dose of 54–60 Gy, over 6 weeks [23,24,25,26]. At clinical progression of the disease, which occurs in almost all cases within 3–8 months, reirradiation can be performed as salvage treatment, to improve survival outcomes and quality of life [1,5]. The improvements and progress in imaging, planning, and delivery of radiotherapy have played a significant role in facilitating the possibility of reirradiation of previously irradiated tissue [27].

In 2012, a case series on patients with DIPG undergoing reirradiation demonstrated the feasibility of combining radiotherapy with chemotherapy to improve symptoms and delay progression with minimal toxicity rates. This study also suggested that patients who are most likely to benefit from reirradiation may be those with prolonged responses to initial therapy and a long interval since initial radiation [28].

In the following years, the largest retrospective matched-cohort analysis on behalf of the SIOP-E-HGG/DIPG working group was published. In this work, authors demonstrated a significant benefit in terms of median overall and symptom improvement in patients with DIPG, responding to upfront radiotherapy and undergoing reirradiation at the first progression. According to SIOP-E recommendations, reirradiation of children with DIPG at the first progression can be considered when eligibility criteria are fulfilled and radionecrosis after upfront radiotherapy is excluded [29,30].

In this setting, due to the limited number of studies, the optimal radiation dose for reirradiation has not yet been determined.

In recent years, the first and only prospective phase I/II study on DIPG reirradiation has investigated three different dose levels: 24 Gy in 12 fractions, 26.4 Gy in 12 fractions, and 30.8 Gy in 14 fractions. This study reported clinical improvement in almost all patients and a preference for the 24 Gy in 12 fractions schedule [31].

Regarding higher doses, a large single institution series on 20 patients investigated a response-based dose escalation approach, demonstrating improvement in survival with acceptable toxicity of reirradiation with a median total dose of 41.4 Gy (range 33.8–43.2 Gy) [32].

Beyond alternative fractionations, charged particle beams with protons have also been investigated. Proton beam radiotherapy (PBRT) is a modern technique able to administer a homogenous dose on the target, simultaneously preserving healthy tissues in the direct neighborhood [30]. This ballistic advantage correlates to the physical properties of protons and, in particular, to the possibility of obtaining a highly collimated beam with minimal lateral scatter and reducing to zero the dose behind the target volume, thanks to the distal dose fall-off of the Bragg Peak phenomenon [33,34].

Although proton therapy is a priority in pediatric low-grade glioma, it may be contraindicated in high-grade gliomas, such as DIPG, because of poor outcomes and a short survival prognosis. Furthermore, uncommon but serious morbidities, including symptomatic brainstem injury after proton beam radiation, have been reported. Even though proton therapy is considered theoretically useful in reducing the radiation dose to healthy brain tissue, studies investigating proton therapy in patients with DIPG are scarce; therefore, this technique is not routinely used [35,36,37,38,39].

Regarding the second reirradiation, only a few works are reported in the literature.

La Madrid et al. reported a case series on two patients with DIPG treated with second reirradiation. In this work, one patient was treated with 54 Gy in 30 fractions at the first irradiation, 30.6 Gy in 17 fractions at the second irradiation, and 21.6 Gy in 12 fractions at the third irradiation (BED10cum = 125.3 Gy, BED3cum = 169.9 Gy, EQD2cum = 104.4 Gy and 102 Gy for α/β = 10 Gy and α/β = 3 Gy, respectively).

The second patient received hypofractionated radiotherapy with 39 Gy in 13 fractions at the first irradiation, and 20 Gy in 10 fractions at both the second and third irradiation (BED10cum = 98.7 Gy, BED3cum = 144.7 Gy, EQD2cum = 82.2 Gy and 86.8 Gy for α/β = 10 Gy and α/β = 3 Gy, respectively). This child received an antiangiogenic regimen after the first irradiation, irinotecan and rapamycin after the second irradiation, and temozolomide concurrent to the third irradiation. In this study, treatment was well tolerated with no irradiation-associated acute toxicity identified. The patients died 4 and 12 months after the second reirradiation [40].

Subsequently, Bergengruen et al. reported a single case of DIPG treated with temozolomide and second reirradiation. In this work, the patient received 54 Gy in 30 fractions at the first irradiation, followed by 36 Gy in 18 fractions at the second irradiation and 21.6 Gy in 12 fractions at the third irradiation. In this study, authors reported an EQD2 (α/β = 2) for brain and brainstem (without repair) of 117.8 Gy and 114.1 Gy (α/β2) (BED10cum = 132.4 Gy, BED3cum = 181.0 Gy, EQD2cum = 110.3 Gy and 108.6 Gy for α/β = 10 Gy and α/β = 3 Gy, respectively). Feasibility and tolerability of the second course of reirradiation have been claimed, with no acute neurological symptoms or radiation-induced toxicities. OS was 24 months after initial diagnosis [41].

A comparison between different schedules and EQD2cum is reported in Table 2.

The necessity to quantify all the above-mentioned values (BED10, BED3, EQD2cum, rate of clinical response, radiological response, and adverse events) arises from literature data suggesting a correlation between brain irradiation and the risk of neurocognitive impairment in a dose–volume dependent manner (a higher cumulative cranial dose predisposes survivors to worse IQ scores), as well as a higher sensibility to radiotherapy-induced necrosis in pediatric patients. More specifically, an analysis published in 2020 suggested that children receiving a cumulative radiotherapy dose of 58.9 Gy or 59.9 Gy at 2 Gy per fraction to any part of the brain, including the brain stem, have an approximate 5% risk of necrosis and a 5% risk of a subsequent IQ < 85 when 10%, 20%, 50%, or 100% of the brain is irradiated to 35.7, 29.1, 22.2, or 18.1 Gy, respectively [42].

In clinical practice, deciding the most suitable treatment approach for children with progressive DIPG can be extremely complex. With the aim of investigating a feasible and effective treatment strategy, we performed a retrospective case series on three children treated at our institution from 2018 to 2022. Even if the results of this study are promising, it is important to acknowledge that this work has some limitations that should be taken into account. First and foremost, the study is performed a posteriori, using information on events that have taken place in the past. Additionally, the sample size is quite small. Lastly, it is not exempt from ethical concerns since the primary goal is just to alleviate symptoms and prolong patients’ survival, without negatively impacting their quality of life. For all these reasons, it is still impossible to conclude which treatment strategy is best. However, every therapy must be tailored to the patient’s characteristics and take into account several factors such as the expected clinical response, the risk of adverse events, the children’s life expectations, and the will of the families.

## 5. Conclusions

The management of children with progressive DIPG represents a difficult ongoing challenge for the radio-oncology community, going beyond our understanding of conventional radiobiology. Based both on literature data and daily practice, it can be deduced that first and second reirradiation, administered together with systemic therapy, may represent a feasible and safe approach, able to increase overall survival and disease-free survival in selected patients and improve quality of life.

Greater results are expected to be achieved in the following years thanks to the development of more advanced radiotherapy methods and new discoveries in molecular biology that will allow us to identify the most adequate systemic therapy.

## Figures and Tables

**Figure 1 brainsci-13-01449-f001:**
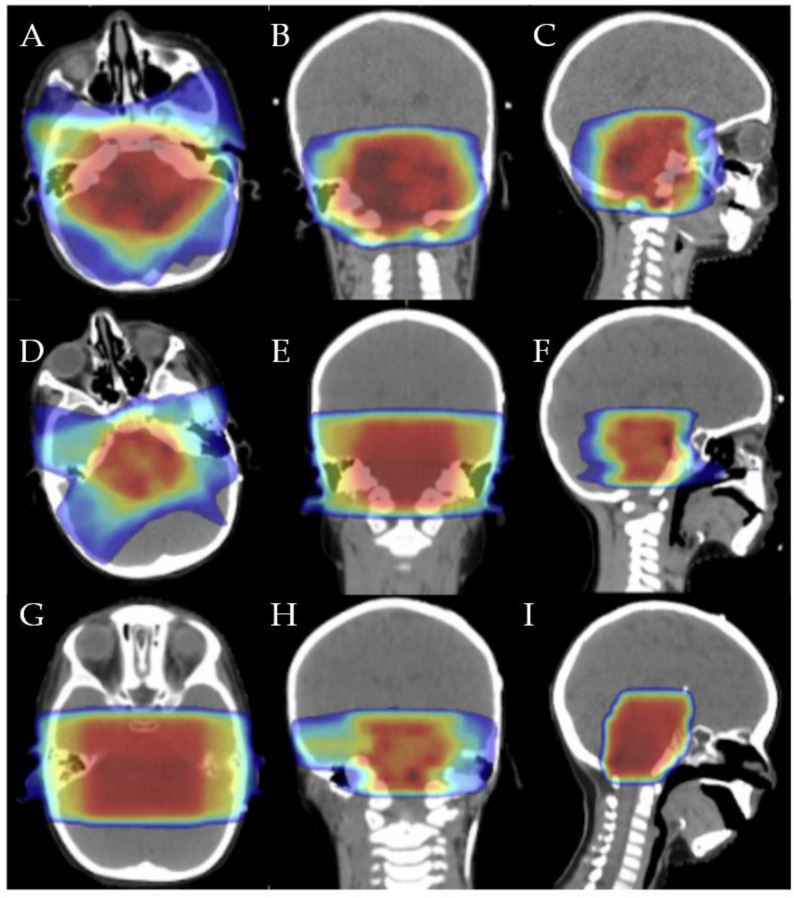
First irradiation isodose distribution in axial (**A**), coronal (**B**), and sagittal (**C**): 20 Gy (dark blue), 32 Gy (light blue), 40 Gy (green), 45 Gy (yellow), 48 Gy (orange), and 54 Gy (red). Second irradiation isodose distribution in axial (**D**), coronal (**E**), and sagittal (**F**): 7.5 Gy (dark blue), 12 Gy (light blue), 15 Gy (green), 17.5 Gy (yellow), 19 Gy (orange), and 22 Gy (red). Third irradiation isodose distribution in axial (**G**), coronal (**H**), and sagittal (**I**): 4 Gy (dark blue), 8 Gy (light blue), 9 Gy (green), 10 Gy (yellow), 11 Gy (orange), and 12 Gy (red).

**Figure 2 brainsci-13-01449-f002:**
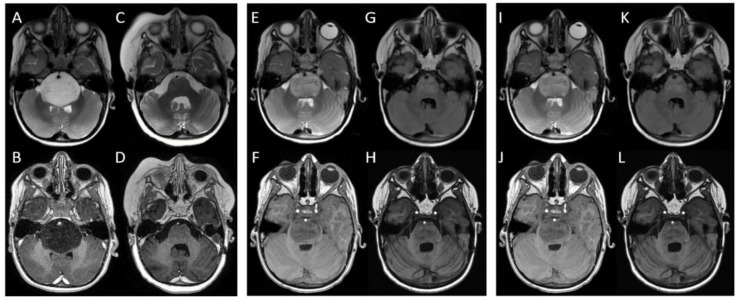
MRI images of a child treated with three courses of radiotherapy: at baseline, the mass located in the pons appeared to protrude and partially obliterate the prepontine cisterns, embrace the basilar artery, displace the cerebellar vermis and middle cerebellar peduncles, and compress the fourth ventricle and Sylvian aqueduct. It appeared to be (**A**) hyperintense in T2-weighted and FLAIR sequences and (**B**) hypointense in T1-weighted sequences. After the first radiotherapy course, the mass decreased, and it was neither obliterating prepontine cisterns nor compressing the fourth ventricle; furthermore, the basilar artery appeared to be free. The mass appeared to be (**C**) less hyperintense in T2-weighted and FLAIR sequences (almost isointense) and (**D**) less hypointense in T1-weighted sequences. At first recurrence (**E**,**F**), the mass was again obliterating prepontine cisterns and imprinting the fourth ventricle. After the second radiotherapy course (**G**,**H**), a considerable reduction in the lesion was seen with all ventricles appearing regular and symmetric. At the second recurrence (**I**,**J**), the lesion appeared to have increased again and extended to the midbrain and posterior cranial fossa, toward cerebellar tonsils. After the third radiotherapy course (**K**,**L**), a partial reduction in the mass was seen.

**Table 1 brainsci-13-01449-t001:** Patients’ time to first and second progression and overall survival.

	Time to First Progression	Time to Second Progression	Overall Survival
Child n. 1	9 months	6 months	24 months
Child n. 2	6 months	4 months	18 months
Child n. 3	6 months	3 months	17 months
Median value ± standard deviation	7 ± 1.7 months	4.3 ± 1.5 months	19.6 ± 3.7 months

**Table 2 brainsci-13-01449-t002:** Comparison between different schedules and EQD2cum in studies administering three courses of radiotherapy.

	I Radiotherapy Course	II Radiotherapy Course	III Radiotherapy Course	EQD2cum (α/β = 10)
Our experience	2 Gy × 27 fx = 54 Gy	1.8 Gy × 11 fx = 19.8 Gy	2 Gy × 6 fx = 12 Gy	85.5 Gy
La Madrid et al. [40]	1.8 Gy × 30 fx = 54 Gy	1.8 Gy × 17 fx = 30.6 Gy	1.8 × 12 fx = 21.6 Gy	104.4 Gy
	3 Gy × 13 fx = 39 Gy	2 Gy × 10 fx = 20 Gy	2 Gy × 10 fx = 20 Gy	82.2 Gy
Bergengruen et al. [41]	1.8 Gy × 30 fx = 54 Gy	2 Gy × 18 fx = 36 Gy	1.8 Gy × 12 fx = 21.6 Gy	110.3 Gy

## Data Availability

The data presented in this study are available on request from the corresponding author. The data are not publicly available due to privacy restrictions.

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
