# Peer review of "The Role of Reirradiation in Childhood Progressive Diffuse Intrinsic Pontine Glioma (DIPG): An Ongoing Challenge beyond Radiobiology"

_brainsci, 2023, doi:10.3390/brainsci13101449_

Round 1
Reviewer 1 Report
Title: Role of Reirradiation in Childhood Progressive Diffuse Intrinsic Pontine Glioma (DIPG): an Ongoing Challenge Beyond Radiobiology
- The title is informative but could be slightly refined for clarity: "The Role of Reirradiation in Childhood Progressive Diffuse Intrinsic Pontine Glioma (DIPG): An Ongoing Challenge Beyond Radiobiology."
Abstract:
- The abstract provides a concise summary of your research, but it could benefit from some restructuring to improve readability. Consider breaking it down into separate sentences or bullet points for each key point.
- Specify the total number of children included in the study.
- The terms "radiobiology" and "vinorelbine and nimotuzumab" should be explained briefly for readers who may not be familiar with these terms.
- Specify the journal's word limit for abstracts, as abstract lengths can vary by journal.
Introduction:
- The introduction is well-structured and clearly states the problem, providing relevant background information.
- Consider providing a more up-to-date reference (if available) for the overall survival rate of DIPG to ensure that your data is current.
Materials and Methods:
- The materials and methods section is informative, but some sentences are long and could be broken down for clarity.
- Specify the institutional review board (IRB) approval for conducting the retrospective study.
- Explain the rationale behind the choice of the total dose of 54 Gy, as it is a crucial aspect of the study.
- Clarify the purpose and methodology of the "Treatment Planning Software (Monaco®)" for target volume definition and dose solutions.
- Mention the criteria used by the local tumor board to determine the eligibility of patients for reirradiation.
- Clarify how "simultaneously preserving healthy tissues in the direct neighborhood" is achieved during proton therapy.
- Provide information on the statistical analysis methods used (e.g., Kaplan-Meier survival analysis, Cox proportional hazards model) if applicable.
Results:
- The results section provides clear data and outcomes of the treatment courses, which is essential.
- Provide detailed information on any adverse events or side effects experienced by the patients during the treatment courses.
- Consider using tables or figures to present data such as survival rates, response rates, and dose calculations.
- Mention the sample size when presenting median values and ranges (e.g., median age 5±1.4 years for how many patients?).
Discussion:
- The discussion is comprehensive and references key studies in the field.
- Include a discussion of the limitations of your study, such as the small sample size and the retrospective nature of the study.
- The abbreviations BED, EQD2, and DVH should be spelled out when first introduced in the discussion.
- Discuss the implications of your findings for clinical practice and future research.
- Clarify the significance of the choice of α/β ratios (10 Gy and 3 Gy) for BED and EQD2 calculations.
- Explain the relevance of IQ <85 in relation to radiation doses in the context of DIPG treatment.
- Consider addressing the potential ethical concerns related to reirradiation in pediatric patients.
- The discussion could benefit from more in-depth exploration of recent developments in the field and potential areas for further research.
Extensive editing of English language required
Author Response
- The title is informative but could be slightly refined for clarity: "The Role of Reirradiation in Childhood Progressive Diffuse Intrinsic Pontine Glioma (DIPG): An Ongoing Challenge Beyond Radiobiology."
Okay, I fixed the title.
Abstract:
- The abstract provides a concise summary of your research, but it could benefit from some restructuring to improve readability. Consider breaking it down into separate sentences or bullet points for each key point.
I tried to make it more readable, unfortunately, I can use only 200 words.
- Specify the total number of children included in the study.
It is already specified in line 18.
- The terms "radiobiology" and "vinorelbine and nimotuzumab" should be explained briefly for readers who may not be familiar with these terms.
Unfortunately, I don’t have enough words to explain these concepts in the abstract section but they will be treated later in the text.
- Specify the journal's word limit for abstracts, as abstract lengths can vary by journal.
Only 200 words.
Introduction:
- The introduction is well-structured and clearly states the problem, providing relevant background information.
Ok, thank you.
- Consider providing a more up-to-date reference (if available)for the overall survival rate of DIPG to ensure that your data is current.
Unfortunately, there are few data available and all of them agree with this value.
Materials and Methods:
- The materials and methods section is informative, but some sentences are long and could be broken down for clarity.
Okay, thank you.
- Specify the institutional review board (IRB)approval for conducting the retrospective study.
Ethical review and approval were waived for this study due to its retrospective, non-interventional nature
- Explain the rationale behind the choice of the total dose of 54 Gy, as it is a crucial aspect of the study.
54 Gy delivered by conventionally fractionated radiotherapy is the standard of care for DIPG, anyway, I added the specifications in lines 74-78.
- Clarify the purpose and methodology of the "Treatment Planning Software (Monaco®)" for target volume definition and dose solutions.
Monaco was the TPS we had at our disposal at that time. Explanations about target volume definition are briefly treated in lines 88-96.
- Mention the criteria used by the local tumor board to determine the eligibility of patients for reirradiation.
I added this mention in lines 100-104.
- Clarify how "simultaneously preserving healthy tissues in the direct neighborhood" is achieved during proton therapy.
This is explained in lines 287-290
- Provide information on the statistical analysis methods used (e.g., Kaplan-Meier survival analysis, Cox proportional hazards model)if applicable.
Since the sample size is very small, we did not perform advanced statistical analysis.
Results:
- The results section provides clear data and outcomes of the treatment courses, which is essential.
Ok, thank you.
- Provide detailed information on any adverse events or side effects experienced by the patients during the treatment courses.
I added this specification in the lines 128-130
- Consider using tables or figures to present data such as survival rates, response rates, and dose calculations.
I created a table to present PFS and OS.
- Mention the sample size when presenting median values and ranges (e.g., median age 5±1.4 years for how many patients?).
I added more mentions about the sample size in the text. Thank you for the suggestion.
Discussion:
- The discussion is comprehensive and references key studies in the field.
Thank you.
- Include a discussion of the limitations of your study, such as the small sample size and the retrospective nature of the study.
I added this specification
- The abbreviations BED, EQD2, and DVH should be spelled out when first introduced in the discussion.
Dose-Volume Histogram (DVH) line 142. 2 Gy Equivalent Normalized Dose (EQD2) lines 147, Biologically Effective Dose (BED) line 118.
- Discuss the implications of your findings for clinical practice and future research.
Ok.
- Clarify the significance of the choice of α/β ratios (10 Gy and 3 Gy)for BED and EQD2 calculations.
I added this specification in lines 121-124
- Explain the relevance of IQ <85 in relation to radiation doses in the context of DIPG treatment.
Briefly explained in line 327.
- Consider addressing the potential ethical concerns related to reirradiation in pediatric patients.
I briefly mentioned it in the discussion, talking about the limitations of the study.
- The discussion could benefit from more in-depth exploration of recent developments in the field and potential areas for further research.
Unfortunately, none of the recent developments has shown significant efficacy, this is why they are not extensively treated but just briefly mentioned.
Thank you for all your suggestions, I also provided editing of the English language in the hope that the work will be more readable.
Reviewer 2 Report
This is not prospective research, but, retrospective case series.
Third patient showed longer survival time, but this is not because of 3rd irradiation. Because 3rd patient lived longer, we could treat 3rd irradiation. Author should discuss this causal relation.
Author Response
This is not prospective research, but, retrospective case series. It is.
- I added the specification in the abstract and material and methods sections to make it clearer.
Third patient showed longer survival time, but this is not because of 3rd irradiation. Because 3rd patient lived longer, we could treat 3rd irradiation. Author should discuss this causal relation
- I added this specification in lines 165-167
Reviewer 3 Report
1. Page 2, line 80- line 81 'Treatment version number is what planning software (Monaco®)?
2. Page 3, line 92 'Please clarify, which linear accelerator is used and vice versa HD or collaborative?
3. Please indicate the dose of dexamethasone (8 mg die).
4. Pages 6, line 217-219 "Reference 19 : [Peereboom, D.M., Shepard, D.R., Ahluwalia, M.S., Brewer, C.J., Agarwal, N., Stevens, G.H., Suh, J. H., Toms, S. A., Vogelbaum, M. A., Weil, R. J., Elson, P., & Barnett, G. H. Phase II trial of erlotinib with temozolomide and radiation in patients with newly diagnosed glioblastoma multiforme. J Neurooncol. 2010, 98(1) 93–9] with a focus on GBM patients receiving erlotinib, temozolomide, and synchronous radiotherapy. Which is not consistent with discussions of PARP1 inhibitors (olaparib, Nilaparib, Veliparib), CDK4/CDK6 inhibitors (PD-0332991), WEE1 kinase inhibitors (MK1775), and angiogenesis inhibitors (bevacizumab).
5. Please describe the limitations of the study in the discussion section.
no
Author Response
Page 2, line 80- line 81 'Treatment version number is what planning software (Monaco®)?
- Version 5.11, I added this specification
Page 3, line 92 'Please clarify, which linear accelerator is used and vice versa HD or collaborative?
- Siemens, Oncor, I added this specification
Please indicate the dose of dexamethasone (8 mg die).
- Ok, done
Pages 6, line 217-219 "Reference 19 : [Peereboom, D.M., Shepard, D.R., Ahluwalia, M.S., Brewer, C.J., Agarwal, N., Stevens, G.H., Suh, J. H., Toms, S. A., Vogelbaum, M. A., Weil, R. J., Elson, P., & Barnett, G. H. Phase II trial of erlotinib with temozolomide and radiation in patients with newly diagnosed glioblastoma multiforme. J Neurooncol. 2010, 98(1) 93–9] with a focus on GBM patients receiving erlotinib, temozolomide, and synchronous radiotherapy. Which is not consistent with discussions of PARP1 inhibitors (olaparib, Nilaparib, Veliparib), CDK4/CDK6 inhibitors (PD-0332991), WEE1 kinase inhibitors (MK1775), and angiogenesis inhibitors (bevacizumab).
- I fixed the references.
5. Please describe the limitations of the study in the discussion section.
- Okay, Done
Round 2
Reviewer 1 Report
Acceptable for publication.
Reviewer 2 Report
I recognized significant improvement of this manuscript.